# Diagnostic Value of Umbilical Cord Blood Interleukin-6 Level in Premature Infants with Early-Onset Sepsis

**DOI:** 10.3390/children12030301

**Published:** 2025-02-27

**Authors:** Jinfang Yuan, Yufeng Wu, Yahui Zhang, Lin Zeng, Jiansuo Zhou, Meihua Piao, Xiaomei Tong, Yuan Wei, Liyan Cui, Tongyan Han

**Affiliations:** 1Department of Pediatric, Peking University Third Hospital, Beijing 100191, China; yuanjf9@126.com (J.Y.); huiyaya326@163.com (Y.Z.); pmh1990@sina.com (M.P.); tongxm2007@126.com (X.T.); 2Clinical Laboratory, Peking University Third Hospital, Beijing 100191, China; wuyufeng0825@163.com (Y.W.); locksword@126.com (J.Z.); cuiliyan@bjmu.edu.cn (L.C.); 3Research Center of Clinical Epidemiology, Peking University Third Hospital, Beijing 100191, China; zlwhy@163.com; 4Department of Obstetrics and Gynecology, Peking University Third Hospital, Beijing 100191, China; weiyuanbysy@163.com

**Keywords:** premature infant, umbilical cord blood, interleukin-6, early-onset sepsis, diagnostic value

## Abstract

Objective: Early-onset sepsis (EOS) is a serious, grave, and frequently fatal condition in premature infants. This study aimed to assess the diagnostic value of interleukin-6 (IL-6) levels in umbilical cord blood for identifying EOS in preterm infants. Methods: This prospective cohort study was conducted on preterm infants between May 2019 and April 2021. Based on the diagnostic criteria for EOS, the participants were divided into EOS and non-EOS groups. Receiver operating characteristic (ROC) curve analysis was performed to evaluate the diagnostic efficacy of cord blood IL-6 levels for EOS. Results: The levels of IL-6 were significantly higher in the EOS group (*n* = 10) compared to the non-EOS group (*n* = 178) [617.5 pg/mL (323.3, 1579.8) vs. 49.7 pg/mL (15.8, 142.8), respectively; *p* = 0.000]. ROC curve analysis demonstrated that a cutoff value of 250.5 pg/mL for cord blood IL-6 yielded a sensitivity of 90%, specificity of 82%, and area under the curve of 0.876, with a confidence interval of 0.753–0.999, indicating its high accuracy as a diagnostic marker for EOS among preterm infants (*p* < 0.001). Conclusions: The detection of IL-6 in the umbilical cord blood offers convenience and exhibits significant diagnostic potential for EOS in preterm infants, thereby providing valuable support for clinical decision-making.

## 1. Introduction

Neonatal early-onset sepsis (EOS), defined as sepsis diagnosed within 72 h of birth, remains a severe and fatal condition among premature infants, particularly those with a gestational age of <32 weeks. The morbidity and mortality rates of EOS increase with decreasing gestational age. The gold standard for diagnosing EOS is positive blood culture [1,2]. Nevertheless, false-negative blood cultures are common, with only 5% of children suspected of having EOS presenting with positive blood cultures and 95% of children with EOS receiving treatment based only on suspected symptoms [3]. The clinical manifestations of EOS are not specific, and most neonates exhibit suspected EOS symptoms within 6 h of birth [4].

Early detection and effective treatment of EOS can enhance clinical outcomes of birth in premature or low-birth-weight infants (LBW). A common clinical approach is to adopt a risk-based strategy with a low threshold, which involves initiating antibiotics immediately after birth in preterm infants exposed to potential hazards. This approach reduced the incidence of EOS but increased the number of non-infected infants exposed to antibiotics [5].

Multicenter surveys indicate that the early antibiotic exposure rate of preterm infants is not consistent with the incidence of EOS and that the exposure rate varies among different medical centers [6]. For each culture-proven EOS case, more newborns were exposed to the potential harm associated with empirical antibiotic treatment [7]. In preterm infants, antibiotic exposure may increase the risk of necrotizing enterocolitis, bronchopulmonary dysplasia, periventricular white matter damage, invasive fungal infections, retinopathy, and death [5]. Early-life microbiota is a crucial factor in normal immune development and long-term health [8], and early-life gut microbiota imbalance is related to childhood or adult disease outcomes, such as atopic dermatitis, asthma, allergic disease, obesity, diabetes, cardiovascular disease, and neurological disorders [9].

The early identification of EOS and the administration of targeted treatments remain a challenge. Fetal inflammatory response syndrome (FIRS) can occur when the fetus is exposed to microorganisms. A higher proportion of newborns with FIRS have EOS compared to those without FIRS who have EOS [10]. Interleukin-6 (IL-6) is significantly elevated in FIRS, and FIRS can be diagnosed by the concentration of IL-6 in umbilical cord blood [11], which is also an early marker of EOS; however, whether elevated IL-6 levels in umbilical cord blood contribute to the early diagnosis of EOS remains to be further explored. Based on perinatal data, this study prospectively analyzed the diagnostic value of umbilical cord blood IL-6 levels in premature infants with EOS.

## 2. Materials and Methods

### 2.1. Subjects

This was a single-center, prospective cohort study. The inclusion criteria were as follows: birth at less than 37 weeks of gestation, delivery to the obstetric department, and immediate transfer to our hospital’s neonatal intensive care unit after delivery between May 2019 and April 2021. Infants with genetic metabolic diseases, nervous system malformations, or other system malformations were excluded.

### 2.2. Methods

#### 2.2.1. EOS Diagnostic Criteria

A confirmed neonatal EOS is defined as a positive blood culture or the presence of two or more of the following clinical signs within 72 h of birth: temperature instability, respiratory symptoms (apnea and respiratory distress), cardiovascular symptoms (hypotension and tachycardia), neurological symptoms (convulsions and hypotonia), or abdominal symptoms (vomiting, poor sucking ability, and bloating), accompanied by ≥2 non-specific tests being positive (white blood cell (WBC) count ≥30 × 10^9^/L or <5 × 10^9^/L; platelet count <100 × 10^9^/L; C-reactive protein (CRP): ≥3 mg/L at 6 h of age, ≥5 mg/L at 6~24 h of age, ≥10 mg/L at 24 h age; procalcitonin (PCT) ≥ 0.5 mg/L) [12].

#### 2.2.2. Collection of Umbilical Cord Blood

Blood samples were obtained from clamped umbilical cords after the neonates were delivered and before the placenta was delivered. A total of 5 mL of umbilical cord blood was extracted with a disposable sterile syringe and transferred to the EDTA anticoagulant tube (Jishuitan Medical Technology Co., LTD, Beijing, China). The tube was gently reversed and centrifuged at 3000 rpm at 4 °C for 10 min immediately to separate the plasma. Serum IL-6 levels in the umbilical blood were determined using a chemiluminescence enzyme immunoassay (Cobas e801; Roche Diagnostics, Mannheim, Germany).

#### 2.2.3. Clinical Data

For every newborn, the data records included (1) prenatal data: maternal age, complications during pregnancy, drug use, prenatal fever, and whether antibiotics were used; (2) intrapartum data: neonatal birth status, sex, gestational age, weight, delivery mode, Apgar score at birth, amniotic fluid, placenta and umbilical cord, placental pathology, and IL-6 detection in umbilical cord blood; and (3) postpartum data: blood routine, CRP, PCT in umbilical cord blood, symptoms after admission, clinical diagnosis, positive blood culture time, strain, and clinical outcome of the child within 24 h of birth.

#### 2.2.4. Statistical Analysis

IBM Statistical Package for Social Sciences software version 25 was used for statistical analysis. Normal distribution was tested for continuous variables. Quartile M (QL and QU) was used for data with a non-normal distribution. The Mann–Whitney nonparametric test was used to compare the differences between the case and control groups. The crosstab test was used to count data and Fisher’s exact test measured low-frequency data. A multivariate binary logistic regression model was used to identify the indicators of diagnostic significance for EOS, and the receiver operating characteristic (ROC) curve was further analyzed to evaluate the diagnostic efficacy of cord blood IL-6 in the EOS of premature infants. Statistical significance was set at *p* < 0.05. Hosmer and Lemeshow tests were used to test the goodness of fit of the regression model to ensure that the model fit the data adequately.

## 3. Results

### 3.1. Demographic and Clinical Characteristics

From May 2019 to April 2021, 188 preterm infants met the inclusion criteria and were categorized into the EOS group (10 cases) and non-EOS group (178 cases) in accordance with the EOS diagnostic criteria. The demographic and clinical characteristics of the patients are shown in Table 1. After undergoing a normal distribution test, it was found that gestational age, birth weight, birth length, birth head circumference, maternal age, and indicators of cord blood inflammation (IL-6, WBC count, PCT, and CRP) did not conform to a normal distribution; thus, nonparametric tests were used. For sex, postpartum asphyxia, maternal pregnancy with diabetes, hypertension, premature rupture of membranes, prenatal use of dexamethasone, mode of delivery, placental pathology of chorioamnionitis, the crosstab test, and Fisher’s exact test were used.

The study population’s median gestational age and birth weight were 32+5 weeks (25+1 to 36+3 weeks) and 1624 g (range, 620–3400 g), respectively. There were no disparities in the demographic characteristics between the groups.

Regarding clinical characteristics, there were no differences in pregnancy complications, rate of premature rupture of membranes (2/10 vs. 27.5%), or placental chorioamnionitis (5/10 vs. 32.6%) among the mothers in the groups. Neonatal asphyxia rates did not differ. The WBC, CRP, and PCT levels in umbilical cord blood were compared, and there were no statistically significant differences. IL-6 levels were significantly elevated in the EOS group (617.5 (323.3, 1579.8) vs. 49.7 (15.8, 142.8) (pg/mL); *p* = 0.000).

Among the ten preterm infants diagnosed with EOS, three died and seven survived. Further comparative analysis demonstrated no statistically significant differences in gestational age, sex, premature rupture of membranes, placental pathology, postnatal asphyxia, IL-6 in the umbilical blood, PCT level, WBC count, and CRP level. All three deaths occurred through vaginal delivery, and all seven surviving cases involved cesarean delivery. There were statistically significant differences in the delivery modes. IL-6 levels in the umbilical blood of the three deceased patients were 344.0 pg/mL, 2623.00 pg/mL, and > 5000.0 pg/mL, which were significantly increased.

### 3.2. ROC Curve Analysis

A binary logistic regression analysis was conducted on maternal age, pregnancy complications, delivery mode, Apgar score at birth, amniotic fluid and placental umbilical cord conditions, placental pathology, gestational age, birth weight, and umbilical cord blood IL-6, WBC, CRP, and PCT levels. Only the umbilical cord IL-6 level was found to have diagnostic significance for EOS (*p* = 0.001, OR = 1.004). ROC curve analysis of umbilical cord IL-6 was performed to assess its diagnostic value in the EOS of preterm infants. The results indicated (Figure 1) that for the diagnosis of EOS in preterm infants, the cutoff value of IL-6 was 250.5 IU/L (sensitivity, 90.0%; specificity, 82.0%; AUC, 0.876; 95%CI, 0.753–0.999; *p* < 0.001), and the positive and negative predictive values were 22.0% and 99.3%, respectively. The Hosmer–Lemeshow goodness-of-fit test was further utilized for verification, and *p* = 0.484 > 0.05, suggesting that the scoring model functioned well.

## 4. Discussion

EOS in preterm infants remains a clinical challenge and is characterized by high morbidity and mortality, a lack of specificity in clinical manifestations, and the absence of early and reliable identification methods. However, initiating antibiotic treatment 12 h after the first clinical symptom fails to prevent the fulminant clinical course of EOS, with septic shock and death [13]. Therefore, it is necessary to develop a reliable approach for the early identification and guidance of clinical decision-making. Consequently, we evaluated the role of cord blood IL-6 in the early diagnosis of EOS.

Our findings indicate that cord blood IL-6 can be detected rapidly and has a significant diagnostic value for EOS. The cutoff value was 250.5 IU/L (sensitivity, 90.0%; specificity, 82.0%). The positive and negative predictive values were 22.0% and 99.3%, respectively.

Numerous biomarkers exist for neonatal sepsis [14]. Hematological (blood cultures) and inflammatory (CRP, PCT, and serum amyloid A) biomarkers are currently used in clinical practice. Some cytokines (IL-6, IL-8, and tumor necrosis factor) and cell adhesion molecules (preseason, soluble triggering receptor, and cluster differentiation molecule-64) may have potential applications in treating neonatal sepsis.

IL-6 is an early marker for neonatal sepsis. After the onset of bacteremia, pathogens activate toll-like receptors (TLRs), leading to the release of pro-inflammatory cytokines by monocytes and macrophages. The potential of cytokine IL-6 lies in its ability to act as early warning signals of infection [15]. It is released within 2-4 h, peaks at the next 6 h, and decreases during the subsequent 24 h [15]. Chiesa et al. investigated the dynamic change trend of IL-6 in 148 healthy infants (113 preterm and 35 near-term infants) within 48 h after birth. The average concentration of IL-6 in healthy full-term neonates was 1.69 pg/mL at birth, 4.09 pg/mL at 24 h, and 3.45 pg/mL at 48 h. The corresponding IL-6 values of healthy near-term infants were 10.9 pg/mL, 9.3 pg/mL, and 8.4 pg/mL, respectively [16]. A recent meta-analysis suggested that IL-6 levels are more accurate in preterm infants. Early specimen collection at the time of suspected sepsis has the highest sensitivity. The diagnostic value of IL-6 in cord blood is higher than in peripheral blood [17].

Another study of 128 neonates who fulfill at least one prenatal risk factor for EOS revealed that cord IL-6 was a better predictor of EOS than CRP, with a cutoff value of 255.87 pg/mL, a sensitivity of 90%, and a specificity of 87.4%. Chorioamnionitis and a 1 min Apgar score were independent risk factors for EOS. IL-6 levels are not significantly correlated with gestational age [18]. In our study, we included all premature babies, and there was no difference in the presence of placental chorioamnionitis or postnatal asphyxia between the two groups. Only the cord blood IL-6 levels were significantly elevated in the EOS group. The cutoff value for the diagnosis of EOS was similar to that in this study. It shows that IL-6 is an effective and stable indicator for predicting EOS. Furthermore, our study suggests that IL-6 can be used to predict EOS in preterm infants regardless of prenatal risk factors for EOS.

A comparison of 67 preterm infants with EOS and 115 infants without EOS showed that the specificity and sensitivity of serum IL-6 levels measured immediately after birth were 72.8% and 75.0%, respectively. The area under the curve was 0.804, and the cutoff value was 40 ng/L. The specificity or sensitivity can increase if IL-6 is combined with another perinatal factor [19]. The cutoff value in this study was lower than the umbilical cord blood IL-6 level in our study. In this study, there was a significant difference in the duration of premature rupture of membrane (PROM) between the two groups, with the control group (52.4 h) having a significantly shorter duration than the case group (139.4 h). In contrast, there was no significant difference in our study. PROM can lead to bacterial colonization of the amniotic fluid, which produces and releases pathogen-related molecular patterns, activates related signaling pathways, and synthesizes IL-6 [20]. Previous studies on preterm infants have shown that cord blood IL-6 levels are significantly increased in preterm infants whose mothers have PROM [21,22,23,24]; PROM is not a single risk factor for EOS. Our study showed no significant difference in the PROM time between the two groups. The difference in the IL-6 cutoff value between the two studies suggests that the predictive value of IL-6 in the diagnosis of EOS should be appropriately strict in cases of PROM. Moreover, as the cytokine is essential in the host response to stress and tissue injury, the elevation of IL-6 in infected neonates at risk of EOS is hindered [17].

Moreover, in a retrospective study of 445 infants, among whom 53 developed EOS, IL-6 levels were monitored twice within 24 h of delivery. The sensitivity and specificity for the presence of EOS of the peak IL-6 level of more than 200 ng/L were 89% and 77%, respectively, indicating that peak IL-6 was reliable in the systemic inflammatory response and helpful in excluding EOS within the first 24 h [25]. In future studies, it is necessary to compare cord blood IL-6 levels and the changing trends of IL-6 levels after birth to guide clinical decision-making.

In a case report of Gram-negative bacilli EOS in preterm infants, six preterm infants with gestational age 23.4–28.2 weeks and birth weights of 570–1080 g had IL-6 levels higher than 1000 pg/mL after birth. Three of them died, and three survived. In all three surviving newborns, the time to peak of serum IL-6 levels was 12 h after birth, and serum IL-6 levels decreased to 100 pg/mL at 72 h after birth, whereas the serum IL-6 levels of the three neonates who passed away increased significantly before death. Correspondingly, the serum CRP levels of the six neonates did not peak within 12 h, indicating that serum IL-6 levels rapidly and significantly changed and were superior to CRP levels in monitoring the severity of sepsis [26]. Changes in white blood cell count were not assessed. In our study, three patients died, and the level of IL-6 in the cord blood increased significantly after birth; however, the change in IL-6 levels after birth was not monitored. In future studies, IL-6 levels should be continuously monitored to determine whether the changing trend of IL-6 can predict the risk of death in infants with EOS.

The retrospective cohort study by Ting, J Y et al. showed that the duration of empirical antibiotic therapy within the first week after birth is associated with increased odds of mortality or any major morbidity (severe neurologic injury, retinopathy of prematurity, necrotizing enterocolitis, chronic lung disease, or hospital-acquired infection) [27]. Other meta-analyses also showed that broad-spectrum antibiotic exposure is associated with an increased risk of invasive fungal infections [28]. In addition to the existing comprehensive consideration of clinical manifestations and culture results, cord blood IL-6 levels provide another factor to assist in clinical decision-making regarding the duration of antibiotic use.

No other factors were found to predict EOS in preterm infants. Among the perinatal parameters and other inflammatory indicators after birth, no other factors were found to be predictive of EOS. It has been shown that under stress conditions, preterm infants have lower CRP expression and higher IL-6 expression compared with term infants. In acute stress, IL-6 levels increase significantly, whereas CRP levels increase significantly in chronic stress. In preterm infants with EOS, IL-6 levels are more likely to increase, whereas CRP levels do not improve significantly [29].

## 5. Strengths and Limitations

In this study, we used a prospective cohort design to study the diagnostic value of inflammatory markers in EOS in preterm births. We effectively compared IL-6 with other inflammatory markers and found IL-6’s superior diagnostic value in EOS prediction. Thanks to the possible use of automation in the IL-6 measurement, more consistent, robust, and definitive results could be obtained for defining a diagnostic cutoff, making it more convenient for clinical application. This study still has some limitations. First, the number of EOS cases was small, and this may have introduced selection bias; this can be further investigated by expanding the sample size. Second, we used a combination of definitive and clinical diagnoses for the diagnosis of EOS, as blood cultures may lead to false-negative results owing to insufficient blood collection in preterm infants, especially in infants with very low birth weight. Third, this study had a single-center design, which may limit the generalizability of the results to other populations and healthcare settings. Furthermore, we did not assess IL-6 levels postnatally, which could provide insights into the progression and severity of EOS.

## 6. Conclusions

Cord blood IL-6 detection is timely and convenient, and it does not increase the autologous blood taken after birth in preterm infants. It is also susceptible and specific for diagnosing EOS in preterm infants and is valuable in assisting clinical decision-making. This study provides preliminary data that need to be confirmed on a larger patient cohort.

## Figures and Tables

**Figure 1 children-12-00301-f001:**
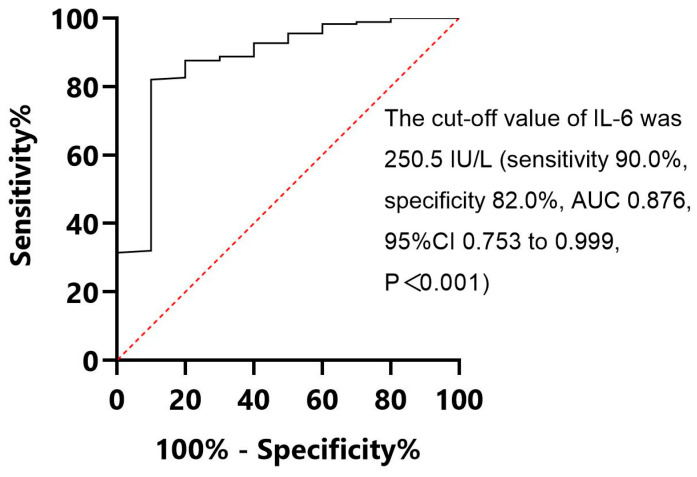
ROC curve of IL-6 in cord blood.

**Table 1 children-12-00301-t001:** Demographic and clinical characteristics.

	EOS Group (*n* = 10)	Non-EOS Group (*n* = 178)	Χ^2^ Value	*p* Value
Gestational age (weeks)	31.0 (28.4, 33.3)	32.0 (30.0, 33.3)	−1.016	0.310
Length (cm)	40.0 (34.0, 43.5)	40.0 (37.0, 43.0)	−0.312	0.755
Birth weight (g)	1820 (1090, 2190)	1610 (1287, 1880)	−0.633	0.527
Head circumference (cm)	29.3 (25.5, 31.1)	29.0 (27.5, 30.6)	−0.090	0.928
Sex (male/female)	3/7	87/91		0.202
Asphyxia (yes/no)	4/6	34/144		0.119
Maternal condition				
Age	36 (32, 37.3)	33 (29, 36)	−1.545	0.122
Gestational diabetes mellitus	3/10	21.3%		0.377
Pregnancy with hypertensive disease	4/10	27.5%	0.472	0.300
Dexamethasone	8/10	90.4%	1.138	0.267
Premature rupture of membranes (PROM)	2/10	27.5%		0.459
Placental chorioamnionitis	5/10	32.6%	1.289	0.256
Method of delivery (vaginal delivery/cesarean section)	3/7	45/133		0.493
Indicators of cord blood inflammation				
IL-6 (pg/mL)	617.5 (323.3, 1579.8)	49.7 (15.8, 142.8)	−3.996	0.000 *
WBC (×10^9^/L)	9.5 (6.1, 14.0)	8.1 (6.0, 12.0)	−0.631	0.540
PCT (ng/mL)	0.4 (0.1, 31.5)	0.2 (0.1, 0.3)	−1.914	0.056
CRP (mg/L)	0.8 (0.5, 18.1)	1.0 (0.5, 1.0)	−0.087	0.931

* *p* < 0.05 was statistically significant.

## Data Availability

The original contributions presented in this study are included in the article. Further inquiries can be directed to the corresponding author.

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
