# Peer review of "Diagnostic Value of Umbilical Cord Blood Interleukin-6 Level in Premature Infants with Early-Onset Sepsis"

_children, 2025, doi:10.3390/children12030301_

Round 1
Reviewer 1 Report
Comments and Suggestions for Authors
The authors adress the potential of IL-6 to diagnose rapidly early onset sepsis in premature infants.
1. One of the main ethical concerns for this study is that there is no statement on ethics committee approval. Also, nothing about written informed consent for the collection of cord blood from the parents of the patients.
2. Second, I woul like to ask which isthe originality of this study, since the authors have found similar results to a previous one:
"Another study of 128 neonates revealed that cord IL-6 was a better predictor of EOS 180 than CRP, with a cutoff value of 255.87 pg/ml, a sensitivity of 90%, and a specificity of 181 87.4%. Chorioamnionitis and a 1-minute Apgar score were independent risk factors for 182 EOS. IL-6 levels are not significantly correlated with gestational age [18]. In our study, 183 there was no difference in the presence of placental chorioamnionitis or postnatal as- 184 phyxia between the two groups. Only the cord blood IL-6 level was a risk factor, and the 185 cutoff value for the diagnosis of EOS was similar to that in this study"
3. Otherwise, the methodolgy is correct and findings are similar to reference No 18. What additional information do the authors of the present study bring?
Author Response
We thank the Editor and you for your valuable comments and the chance to publish our manuscript. We have modified the manuscript in accordance with your remarks and the editorial policies. We have addressed each point raised in our revised manuscript.

Reviewer 2 Report
Comments and Suggestions for Authors
The present manuscript "Diagnostic value of umbilical cord blood interleukin-6 level in 2 premature infants with early-onset sepsis" of Yuan Jinfang and coworkers presents a clinical study undelining the role of IL-6 as a diagnostic marker for early-onset sepsis.
The study design is appropriate and the results as well as the interpretations are convincing. Since the repertoire of markers to diagnose sepsis is still dissatisfying, the present manuscript is very useful for clinicians. The quantificiation of IL-6 as a sepsis marker is frequently discussed and a matter of publications. In this regard, the study accurates the debate. The discussion section of the manuscript cites a number of candidate markers, which are in focus to diagnose sepsis. However, none of them achieved the relaibility of IL-6.
Taken together, I support the publication of the manuscript in the present form in your esteemed journal.
Best regards
Author Response

(The authors gave the same response as above.)

Reviewer 3 Report
Comments and Suggestions for Authors
1 The authors of the article “Diagnostic value of umbilical cord blood interleukin-6 level in premature infants with early-onset sepsis” have assessed diagnostic value of interleukin-6 (IL-6) levels in umbilical cord blood of preterm infants suffering from early onset sepsis (EOS). They have found that the levels of IL-6 were significantly higher in the EOS group (n=10) compared to the non-EOS group (n=178).
The study presents interesting and important evaluation of IL-6 diagnostic value in EOS. However, the idea is not new, as IL-6 levels in the umbilical cord were found to have diagnostic significance for EOS in many previous studies. In addition, the group of infants with EOS is very small in this study, and statistical power is mainly based on almost 20x bigger non-EOS group.
My other comments and questions are listed below!
1) In the preview of the paper it’s written: Children 2023, 10, x. https://doi.org/10.3390/xxxxx.This prospective study was conducted on preterm infants between May 2019 and April 2021. Was the paper submitted in 2023, or it’s just an error?? If it’s an error, why the authors waited that long to submit the results?
2 2) My major concern for the study is small number of infants with EOS included and presented in the study. Having more cases of infants with EOS, i.e. prolonging the study, would give more power and the study would be more impactful.
3) Why the authors limited their focus on IL-6 only? It is true that IL-6 is more often measured as a predictor of EOS outcome in infants, but also other inflammatory markers might play a role, such are IL-1, IL-8, Tumor Necrosis Factor (TNF-a). Alternatively, maybe use of combination of two or three of them would give a better prediction of preterm infant EOS outcome, as was mentioned in narrative review paper (Stocker & Giannoni, Clin Microbiol Infect 2024;30:22): “inflammatory markers included in algorithms may be a game changer reducing bias and noise in the decision-making process”. Is there any distinction that could be made between factors involved in neonatal EOS and EOS in premature infants? What would be specific pathological role of IL-6 in the onset of EOS?
4) The authors point out the fact that antibiotic exposure of preterm infants might have serious consequences that include death. How early diagnosis and known IL-6 levels would change this approach? Please discuss!
Author Response

(The authors gave the same response as above.)

Reviewer 4 Report
Comments and Suggestions for Authors
The study is devoted to an urgent problem - the comparison and selection of the optimal biomarker for the early diagnosis of sepsis in premature infants. The diagnostic significance of IL-6 in its determination in umbilical cord blood during preterm birth has been convincingly proven. Despite the fact that research in this area is conducted in many countries, this article is very useful and contains important results for neonatology. I would especially like to note the high quality of the discussion, the detailed and interesting analysis of previous studies and reviews on this issue. The practical and scientific significance is beyond doubt, and the article is recommended for publication.
I would like to wish for the authors: In the future, with an increase in the number of observations, pay special attention to another important biomarker - Procalcitonin (PCT). In this work, a large interquartile range is obtained(0.1; 31.5)and the lack of reliability (P value 0,056), but it is safe to assume that monitoring PCT in the blood of premature infants will be useful in the first days of life, when IL6 is already decreasing.
Author Response

(The authors gave the same response as above.)

Reviewer 5 Report
Comments and Suggestions for Authors
REVISION
The aim of this paper was to assess the diagnostic value of interleukin‐6 (IL‐6) levels in umbilical cord blood in preterm infants with Early onset sepsis EOS . It is a prospective study conducted between May 2019 and April 2021, based on the diagnostic criteria for EOS.
The results presented seem to show an excellent performance of the IL‐6 assay in identifying EOS. EOS is a severe and fatal condition among premature infants, with high mortality rates and with challenges in early diagnosis, also considering the high number of cases with false‐negative blood cultures and the risk of empirical antibiotic treatment. So finding an accurate and reliable biomarker that can make early diagnosis would be desirable.
The work is interesting and well developed, but the number of EOS cases considered in the present study remains too limited to draw any definitive conclusions, especially in order to establish a diagnostic cut‐off value.
Some minor revisions
Methods:
‐ Line 80: the term “collection” is incorrectly repeated.
‐ Line 84: Why were the samples centrifuged at 4°C?
‐ Line 86: Please delete the term “software”.
‐ Line 93: Please specify postpartum data characteristics such as sample type (umbilical cord blood or peripheral blood), sampling time and whether PCT, CRP and routine blood tests refer to the mother or the child.
‐ Line 105: Please delete the term “significant”.
Results:
‐ Line 127: Please specify sample type (umbilical cord blood or peripheral blood) used for evaluating WBC, PCT and CRP levels.
‐ Line 127: Please delete “indicators of umbilical cord inflammation”, since altered WBC, PCT and CRP levels may not be necessarily due to umbilical cord localized inflammation
‐ Line 129: Please indicate the unity of measurement.
‐ Lines 143 – 151: It would be appropriate to move these lines before the ROC curve analysis section.
Discussion:
‐ Line 165: Please delete ”and routine blood tests”
‐ Line 191: “The cut-off value in this study….”
This sentence should be modified, because it is not correct to compare IL 6 serum cut‐off with the one obtained in the present study , from umbilical cord blood
‐ Line 231: Please delete or complete “Patients with EOS experience acute stress”.
- Line 234: please you should add this sentence:
“Thanks to the possible use of automation in the IL6 measurement, more consistent, robust and definitive results could be obtained for defining a diagnostic cut-off”
Conclusions: The authors should include in the conclusion section, that this study provides preliminary data which need to be confirmed on a larger patient cohort.
Author Response

(The authors gave the same response as above.)

Round 2
Reviewer 3 Report
Comments and Suggestions for Authors
The authors clarified many points and improved the paper considerably. Most importantly, they explained their focus on IL-6 in EOS. They have added that a higher proportion of newborns with fetal inflammatory response syndrome (FIRS) have EOS compared to those without FIRS having EOS and that FIRS can be diagnosed by the concentration of IL-6 in umbilical cord blood. Nevertheless, they plan to continue the observations and focus on more pro-inflammatory markers. Therefore, I have no further comments or questions for the authors.
Comments on the Quality of English LanguageEnglish is good, just minor adjustments are needed!
Author Response
Dear Reviewer,
Thank you for your positive feedback on the English language quality of our manuscript. We appreciate your comment that only minor adjustments are needed. We have carefully reviewed the manuscript and corrected minor grammatical errors and improved sentence structure for better clarity. We hope the revised version meets your expectations.
Thank you again and look forward to the follow-up processing of the paper.
Sincerely,
Jinfang Yuan
